**Data Availability Statement:** The data used in this study are from the Ethiopia Demographic, and

# Micronutrient intake status and associated factors among children aged 6–23 months in the emerging regions of Ethiopia: A multilevel analysis of the 2016 Ethiopia demographic and health survey

**Tsegaye Gebremedhin***, **Andualem Yalew Aschalew, Chalie Tadie Tsehay, Endalkachew Dellie, Asmamaw Atnafu**

Department of Health Systems and Policy, Institute of Public Health, College of Medicine and Health Sciences, University of Gondar, Gondar, Ethiopia

* tsegishg27@gmail.com

## Abstract

### Background

Micronutrient (MN) deficiency among children is recognised as a major public health problem in Ethiopia. The scarcity of MNs in Ethiopia, particularly in pastoral communities, might be severe due to poor diets mitigated by poor healthcare access, drought, and poverty. To reduce MNs deficiency, foods rich in vitamin A (VA) and iron were promoted and programs like multiple micronutrient powder (MNP), iron and vitamin A supplements (VAS) and or deworming have been implemented. Nationally for children aged 6–23 months, consumption of four or more food groups from diet rich in iron and VA within the previous 24 hours, MNP and iron supplementation within seven days, and VAS and >75% of deworming within the last 6 months is recommend; however, empirical evidence is scarce. Therefore, this study aimed to assess the recommended MN intake status of children aged 6–23 months in the emerging regions of Ethiopia.

### Methods

Data from the Ethiopia Demographic and Health Survey 2016 were used. A two-stage stratified sampling technique was used to identify 1009 children aged 6–23 months. MN intake status was assessed using six options: food rich in VA or iron consumed within the previous 24 hours, MNP or iron supplementation with the previous seven days, VAS or deworming within six months. A multilevel mixed-effect logistic regression analysis was computed, and a p-value of < 0.05 and Adjusted Odds Ratio (AOR) with 95% Confidence Interval (CI) were used to identify the individual and community-level factors.

Health Survey 2016 and can be requested from the MEASURE DHS available at https://www.dhsprogram.com/Data using the details in the Materials and methods section of the paper.

**Funding:** The authors received no specific funding for this work.

**Competing interests:** The authors have declared that no competing interests exist.

**Abbreviations:** ANC, Antenatal Care; AOR, Adjusted Odds Ratio; CI, Confidence Interval; COR, Crude Odds Ratio; CSA, Central Statistical Agency; EA, Enumeration Areas; EDHS, Ethiopian Demographic and Health Survey; FAO, Food and Agriculture Organizations; FMoH, Federal Ministry of Health; ICC, Intra-class Correlation Coefficients; MDD, Minimum Diversified Diet; MN, Micronutrients; PNC, Postnatal Care; VA, Vitamin A; VAS, Vitamin A supplements; WHO, World Health Organization.

## Results

In this analysis, 37.3% (95% CI: 34.3–40.3) of children aged 6–23 months had not received any to the recommended MNs sources. The recommended MNs resulted; VAS (47.2%), iron supplementation (6.0%), diet rich in VA (27.7%), diet rich in iron (15.6%), MNP (7.5%), and deworming (7.1%). Antenatal care visit (AOR: 1.9, 95% CI: 1.4–2.8), work in the agriculture (AOR: 2.2, 95% CI: 1.3–3.8) and children aged 13 to 23 months (AOR: 1.7, 95% CI: 1.2–2.4) were the individual-level factors and also Benishangul (AOR: 2.2, 95% CI: 1.3–4.9) and Gambella regions (AOR: 1.9, 95% CI: 1.0–3.4) were the community-level factors that increased micronutrient intake whereas residence in rural (AOR: 0.4, 95% CI: 0.1–0.9) was the community-level factors that decrease micronutrient intake.

## Conclusions

Micronutrient intake among children aged 6–23 months in the pastoral community was low when compared to the national recommendation. After adjusting for individual and community level factors, women's occupational status, child's age, antenatal visits for recent pregnancy, residence and region were significantly associated with the MN intake status among children aged 6–23 months.

## Introduction

Micronutrient (MN) deficiency among children is recognised as a global public health problem, and it is worse in low- and middle-income countries (LMICs), particularly in Ethiopia [1–3].

The essential MNs needed for life include iron, zinc, calcium, iodine, manganese, chromium, copper, fluoride, and vitamins [4,5]. Although MNs are only needed in small quantities, their absence from diet negatively affects children's survival and development. Furthermore, MN deficiency contribute to debilitating consequences, like stunting, wasting, weak immunity, and delay in cognitive development [6–10]. Notably, MNs are critical during the first 1000 days of a child's life; adequate nutrition during this period promotes healthy growth and development, but less attention has been given to MN [11,12].

According to the United Nations Children's Fund 2019 report, around 340 million children worldwide suffered from hidden hunger caused by MN deficiencies [13]. The problem is much higher in LMICs, and few empirical studies showed that in 2018, only 29% of children aged 6–23 months were fed the minimum diversified diet (MDD) in Ethiopia [14]. 'The MDD score for children 6–23 months old is a population-level indicator designed by the World Health Organization to assess diet diversity as part of infant and young child feeding practices among children 6–23 months old'. Accordingly, the national recommendation is that consumption of four or more food groups from the seven food groups, namely: grains, roots and tubers; legumes and nuts; dairy products; flesh foods (meat, fish, poultry and organ meats); eggs; vitamin A (VA) rich fruits and vegetables; other fruits and vegetables within the previous 24 hours [15].

Similarly, the deficiency of crucial MN are among the significant public health problems in Ethiopia. These deficiencies result from diets with limited diversity, minimal bioavailability, frequent meal skipping, limited access to micronutrient-rich and fortified foods, and low vegetable and fruit intake [16–18]. To prevent MN deficiencies among children in Ethiopia, the

national nutritional supplementation program has been provided in the form of food and supplementation. The recommended MNs for children older than six months include foods rich in VA, foods rich in iron, multiple micronutrient powder, iron and vitamin A supplements (VAS) and or deworming (children older than 12 months) [19–21]. VAS and deworming have been provided for children aged 6 to 59 months semi-annually as a national nutrition program. Nationally for children aged 6–23 months, consumption of four or more food groups from diet rich in iron and VA within the previous 24 hours, MNP and iron supplementation within seven days, and VAS and >75% of deworming within the last 6 months is recommend. Interventions to improve maternal nutrition include multiple micronutrient supplements, food fortification, supplementary food, nutrition education, and counselling, majorly in the community-based nutrition program in Ethiopia [22].

According to the Ethiopia Demographic and Health Survey (EDHS) 2016 report, only 14% of children 6–23 months were received MDD [23]. In addition, the Ethiopian national nutritional supplementation survey (2016) indicated that VAS coverage among children was 63%, which is lower than the national target (more than 90%) [24] and the national prevalence rate of subclinical VA deficiency (serum retinol < 0.7 μmol/L) was severe (37.7%) [25,26].

MN intake is associated with various factors at individual and community levels, including mothers' sociodemographics and child characteristics, dietary habits, community-level lifestyle, and place of residence [27–29]. In addition to the above factors, the use of maternal healthcare services, such as antenatal care (ANC), institutional delivery and postnatal care (PNC), are also associated with the MN intake status of children [30,31].

Although there is documented evidence of insufficient MN intake for agrarian communities and urban dwellers in Ethiopia [14,32], but there is little evidence on MN intake among children aged 6–23 months in emerging regions (Afar, Somali, Benishangul, and Gambela) of Ethiopia where pastoralist communities, with poor cultivation of fruits and vegetables, mainly reside [33]. Additionally, these regions have been identified as the hotspots in the country with high food insecurity, high child malnutrition rates, and recurrent droughts [34,35].

These areas have limited access to health facilities, poor infrastructure, and inaccessible health services [36,37]. However, studies that show the individual and community-level factors associated with MN intake among children are rare. Thus, this study aimed to assess the MN intake status and related factors among children aged 6–23 months in the emerging regions of Ethiopia using the 2016 EDHS data. The findings could give important insights to develop contextual strategies for the mitigation of the problems.

## Materials and methods

### Study settings and data source

The study used the EDHS 2016 data, a nationally representative household survey data collected every five years. It has been implemented by the Central Statistical Agency (CSA) [23] with the primary objective of providing up-to-date estimates of key demographic and health indicators. Administratively, Ethiopia is divided into nine regions (Tigray, Afar, Amhara, Oromia, Benishangul-Gumuz, Gambela, South Nation Nationalities and Peoples' Region (SNNPR), Harari and Somali) and two administrative cities (Addis Ababa and Dire-Dawa) (Fig 1). These regions are again categorised as developed and emerging regions. The emerging regions are Afar, Somali, Benishangul, and Gambela, where scattered pastoralists predominantly live. Inadequate infrastructure, inaccessibility of health services, drought, poverty and absence of clear and detailed regulations are the common characteristics in emerging regions [36,37]. The developed regions are Amhara, Oromia, Tigray, SNNPR and Harari regions and

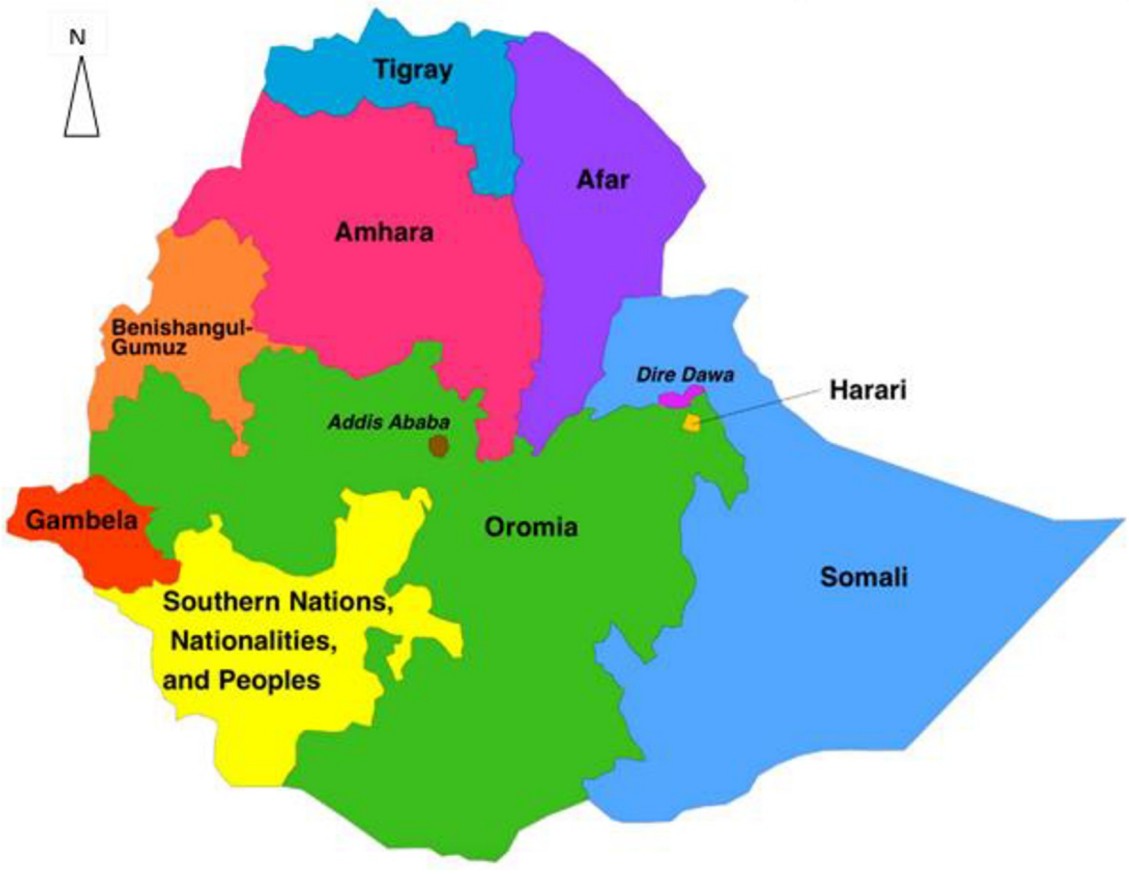

**Fig 1. Map of the study area (Source, CSA:2013).**

the city administrations characterised by a relatively denser population and better infrastructure, and access to health and education services.

## Sampling procedures

The sampling frame for the 2016 EDHS used the 2007 Ethiopian population and housing census, which was conducted by the CSA of Ethiopia. The census used a complete list of 84,915 enumeration areas (EAs), which contains the location, type of residence, and the estimated number of residential households. The 2016 EDHS sample was stratified in two stages, and samples of EAs were selected independently from each stratum. The regions were stratified into urban and rural areas. At each lower administrative level, implicit stratification and proportional allocation were achieved within each sampling stratum before sample selection at different levels.

In the first stage, 645 EAs were selected with probability proportional to the EA size, and each sampling stratum was selected from the given samples. The total residential households in the EA were the EA size, and a household listing operation was implemented. Then, the resulting lists of households were used as the sampling frame for selecting households in the second stage.

Twenty-eight households from each cluster were selected with an equal probability in the second stage, a systematic selection from the newly created household listing. The survey

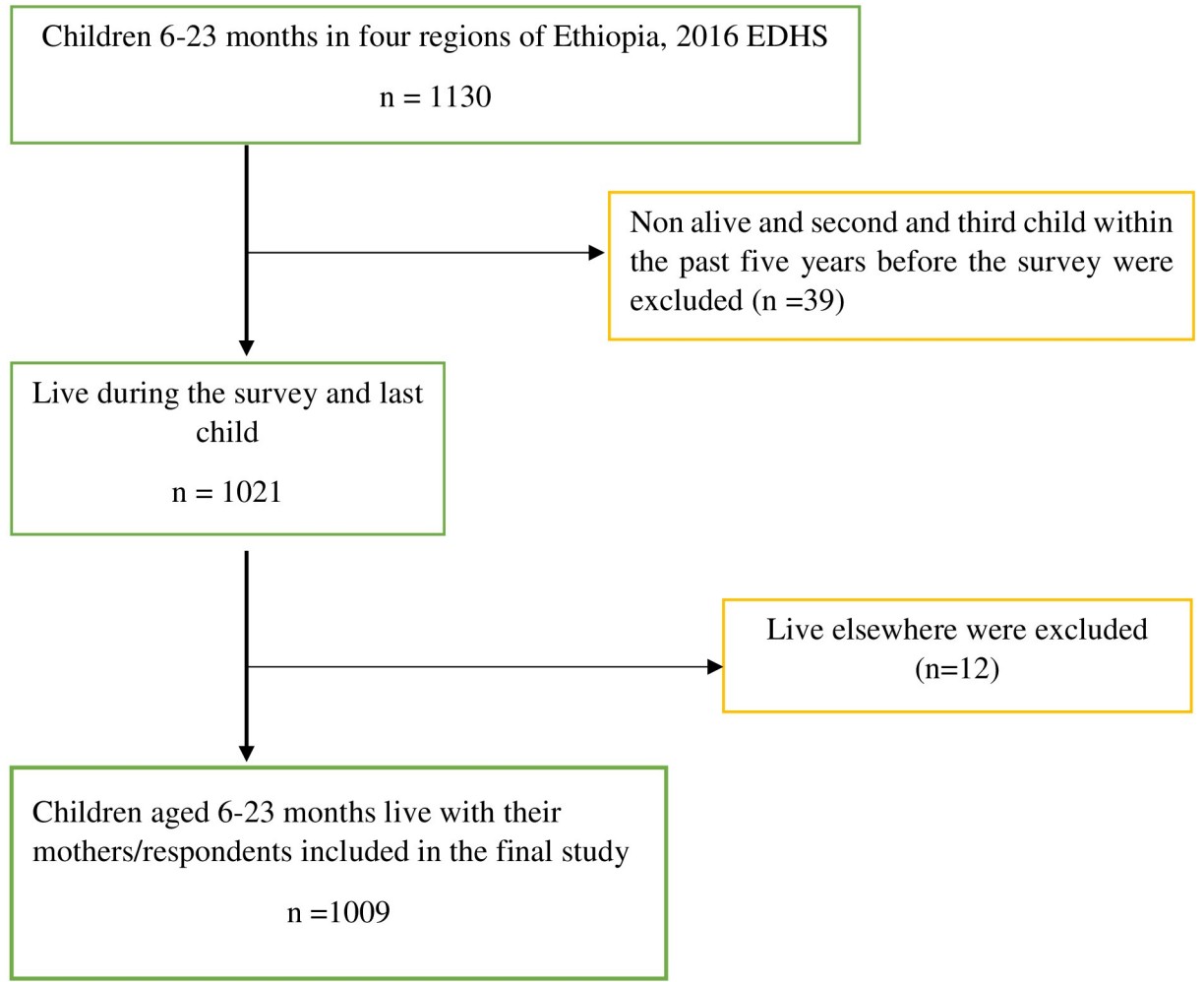

**Fig 2. Sample study selection of children age 6–23 months in emerging regions, EDHS 2016.**

interviewer interviewed only pre-selected households. No replacements or changes of the pre-selected households were allowed in the implementing stages to prevent bias. In this study, the 2016 EDHS childhood datasets of the four emerging regional states: Afar, Benishangul, Gambella and Somali, were used for analysis.

All women aged 15–49 years who are the usual members of the selected households were eligible for the survey. Children aged 6–23 months were the source population and included 1009 mothers/caregivers and their recent children aged 6–23 months in the analysis. In contrast, the second and third child within the last five years (for those who have more than a child), children living with other than their mothers/caregivers were excluded from the analysis (Fig 2). For mothers/caregivers with twins, only one was selected by convenience. Potential individual and community level independent variables were also selected, and further analysis was done.

## Measurements of variables

The dependent variable of the study was MN intake status among children aged 6–23 months, which was determined by respondents' reports and assessment of intake status. So, there were six options: food rich in VA or iron in the last 24 hours, MNP or iron supplement consumed

within the previous seven days, VAS or deworming within the previous six months [19–21,38]. Accordingly, if the respondent reported that the child had eaten' at least one of the minimum recommended MNs, we considered it "Yes"; if the children received none of the minimum recommended MNs, it was considered as "No".

Foods rich in VA were measured by the seven food groups' consumption within the preceding 24 hours. These food groups were I. Eggs, ii. Meat (beef, pork, lamb, chicken), iii. pumpkin, carrots, and squash, iv. any dark green leafy vegetables, v. mangoes, papayas, and others with VA fruits, vi. Liver, heart, and other organs and vii. Fish or shellfish. Accordingly, if the respondent reported that the child had eaten' at least one of these, we considered "yes"; otherwise "no" VA rich food.

Foods rich in iron were measured by the four iron-rich food groups' consumption within the past 24 hours. These groups were i. eggs, ii. meat (beef, pork, lamb, chicken), iii. Liver, heart, and other organs, and iv. Fish or shellfish. Thus, if the respondent reported that the child had eaten' at least one of these, we considered "yes"; otherwise "no" iron-rich food.

Multiple MN powders were assessed by asking the respondents whether their child had received micronutrient powders in the previous seven days.

Iron supplementation was assessed by asking the respondents whether their child had iron supplementation defined as iron pills, sprinkles with iron, or iron syrup in the previous seven days.

VAS and deworming were assessed for those 6–23 months of children whether they received for the last six months or not by reviewing the integrated child health card, which consists of immunisation and growth monitoring history and also from the mother's verbal response.

The obstetric characteristics of women included current pregnancy status and use of maternal health services (ANC, institutional delivery and PNC). The child characteristics include birth weight, and current age. Birth weight was categorised as small, average or large.

The household wealth index was calculated as an index based on consumer goods such as television, bicycle, or car. Household characteristics such as the material used for floor and roof and toilet facilities were also considered in calculating the household wealth index. The household wealth index was computed using principal component analysis and ranked into poor, middle, and rich. Simultaneously, the community-level variables were residence, region, community-level wealth quantile, community-level media exposure, and distance to the nearest health facility.

Community-level wealth quantile was assessed using the asset index based on data from the entire country sample on separate scores prepared for rural and urban households, and combined to produce an index for all households as the community level and ranked into five (poorest, poorer, middle, richer, and richest). In other words, the community level wealth quantile was used to measure the community level poverty and it is a relative measure of how wealth is distributed within the population from the quantiles were calculated.

Community media exposure was assessed as "yes" if they have access to all three media (newsletter, radio, and television) at least once a week, otherwise "no" if they did not have any media exposure.

Distance to the health facility was assessed by the question "distance to the nearest health facility is a problem?" and the responses were categorised as "big problem" or "not a problem" [39].

## Data processing and statistical analysis

The data were cleaned, re-coded and analysed using STATA (StataCorp, College Station, TX) version 14. Descriptive statistics were presented using tables and narration to describe the

magnitude of MN intake status by sociodemographic, maternal obstetric and child characteristics.

A multilevel analysis was conducted after checking the eligibility. The model eligibility was assessed by calculating the Intra-class Correlation Coefficient (ICC) and a model with ICC greater than 10% for multilevel analysis. In this study, the ICC was 27.3%. Since the data were hierarchical (individuals were nested within communities), a two-level mixed-effects logistic regression model was fitted to estimate both the individual and community level variables (fixed and random effect) on MN intake status, and the log of the probability of MN intake was modelled using the formula as follows [40]:

$$log\left[\frac{\pi ij}{1 - \pi ij}\right] = \beta_o + \beta_1 X_{ij} + \beta_2 Z_{ij} + U_j + e_{ij}$$

Where i is an individual level unit and j is a community-level unit; X and Z refer to individual and community-level variables, respectively; $\pi ij$ is the probability of MN intake for the i[th] child in the j[th] community; the β's are the fixed coefficients. Whereas $\beta_0$ is the intercept; the effect on the probability of MN intake in the absence of influence of predictors, and $u_j$ showed the community's effect (random effect) on MN intake for the j[th] community and $e_{ij}$ showed random errors at the individual levels. By assuming each community had different intercepts (β0 + Uj) and fixed coefficient (β1,2), the clustered data nature and the within and between community variations were considered.

Bivariable and multivariable analyses were computed. In the bivariable logistic regression analysis, a p-value of less than 0.2 was used to fit three models (models for the individual, community, and individual and community levels). Then, in the final model (fixed effect), a p-value of less than 0.05 and an Adjusted Odds Ratio (AOR) with a 95% confidence interval (CI) were used to estimate the association of individual and community level factors with MN intake status.

The measures of variation (random-effects) between clusters were reported using ICC and proportional change in variance (PCV). The ICC refers to the ratio of cluster variance to total variance, and it tells us the proportion of the total variance in the outcome variable that is accounted at the cluster level. The loglikelihood test was used to estimate the goodness of fit of the final adjusted model compared to the preceding models. A model with the smallest value of loglikelihood is better; accordingly, model three (a model for both individual and community-level variables) had the lowest value.

### Ethical considerations

The ethical approval and permission to access the data were obtained from the MEASURE DHS (available from https://www.dhsprogram.com/Data/: accessed on April 06, 2020) after a brief study concept was submitted.

## Results

### Sociodemographic and economic characteristics of participants

Table 1 shows the sociodemographic and economic characteristics of the study participants. A total of 1009 mothers/caregivers with children aged 6–23 months were included in the final analysis. The mothers' mean age was 27.5 (SD ± 6.3) years; the majority (72.4%) of the households were in the poor wealth index; the mean family size was 5.9 (SD± 2.3). Religious preference for 71.1% of the mothers was Muslim.

**Table 1. Sociodemographic and economic characteristics of study participants in emerging regions of Ethiopia, 2016 (n = 1009).**

| Variables | Category | Frequency (n) | Percent (%) |
|---|---|---:|---:|
| Age of mothers/caregivers in years | 15–24 | 341 | 33.8 |
| | 25–34 | 497 | 49.3 |
| | >=35 | 171 | 16.9 |
| Religion | Muslim | 717 | 71.1 |
| | Protestant | 173 | 17.1 |
| | Orthodox | 85 | 8.4 |
| | Others* | 34 | 3.4 |
| Sex of household head | Male | 674 | 66.8 |
| | Female | 335 | 33.2 |
| Household wealth index | Poor | 730 | 72.4 |
| | Middle | 78 | 7.7 |
| | Rich | 201 | 19.9 |
| Current marital status | Married | 961 | 95.2 |
| | Unmarried | 48 | 4.8 |
| Educational status of mothers/caregiver's | No education | 715 | 70.9 |
| | Primary education | 199 | 19.7 |
| | Secondary education | 67 | 6.6 |
| | Higher | 28 | 2.8 |
| Educational status of husband's/partner's (n = 961) | No education | 570 | 59.3 |
| | Primary education | 197 | 20.5 |
| | Secondary education | 104 | 10.8 |
| | Higher | 90 | 9.4 |
| Mother's/caregiver's occupation | No work | 671 | 66.5 |
| | Professional worker | 80 | 7.9 |
| | Agricultural worker | 189 | 18.7 |
| | Others** | 69 | 6.8 |
| Husband's/partner's occupation (n = 961) | No work | 139 | 14.5 |
| | Professional worker | 175 | 18.2 |
| | Agricultural worker | 448 | 46.6 |
| | Others** | 199 | 20.7 |

*Catholic, traditional, Joba.

**Daily labor, merchant.

**Obstetric characteristics of mothers/caregivers.** The obstetric characteristics of mothers/caregivers are presented in Table 2. Of the total women, 56.4% of women had ANC; 26.2% delivered at health facilities, and 7.2% of them had PNC checks within two months after delivery.

**Child characteristics and common childhood illness.** Table 3 shows the child characteristics and common childhood illnesses. Of the total children, 42.0% of them had average birth weight, and 15.9% had diarrhoea within the last two weeks.

**Community-level variables.** The majority (83.9%) of the participants were rural dwellers; 63.1% are in the poorest wealth quantile (Table 4).

**Micronutrient intake status.** Overall, 37.3% (95% CI: 34.3–40.3) of children aged 6–23 months had not received any to the recommended MNs sources. Only 27.8% (95% CI: 25.0–30.5) of the children consumed foods rich in VA within the previous 24 hours and 15.6% (95% CI: 13.3–17.8) consumed foods rich in iron within the previous 24 hours; 7.5% (95% CI: 5.9–

**Table 2. Obstetric characteristics of participants in the emerging regions of Ethiopia, 2016 (n = 1009).**

| Variables | Category | Frequency (n) | Percent (%) |
|---|---|---|---|
| ANC visit | Yes | 569 | 56.4 |
| | No | 440 | 43.6 |
| Desire for more children | Wants | 798 | 79.1 |
| | Undecided | 43 | 4.3 |
| | Wants no more | 168 | 16.6 |
| Place of delivery | Home | 745 | 73.8 |
| | Health facility | 264 | 26.2 |
| PNC check up | Yes | 73 | 7.2 |
| | No | 936 | 92.8 |
| Current pregnancy status | Pregnant | 90 | 8.9 |
| | Non-pregnant | 919 | 91.1 |

ANC: Antenatal care.

PNC: Postnatal care.

9.2) received multiple MNP within the last seven days; 6.0% (95% CI: 4.5–7.4) received iron supplements, and 47% (95% CI: 44.1–50.3) of the children received VAS within the previous six months (Table 5).

## Random effects (measures of variation)

There was a significant variation in the intake of MNs among children aged 6–23 months across the communities (clusters). The intra-cluster correlation coefficient (ICC) in the null model (model 0) for MN intake was 0.273. In other words, 27.3% of the variation in MN intake among children aged 6–23 months is due to the differences between regions/clusters (between-cluster variation) (Table 6).

**Individual and community-level factors of micronutrient intake status (fixed effects).** In the final model (model 3), after adjusting for individual and community level factors, women's occupational status, child's age, ANC for current pregnancy, residence and region were significantly associated with the MN intake status among children aged 6–23 months. But, mothers' educational status, being head of household, mothers' occupation, household wealth

**Table 3. Child characteristics and common childhood illness among children aged 6–23 months in the emerging regions of Ethiopia, 2016 (n = 1009).**

| Variables | Category | Frequency (n) | Percent (%) |
|---|---|---|---|
| Current age of the child (months) | 6–12 | 462 | 45.8 |
| | 13–23 | 547 | 54.2 |
| Child's birth weight | Large | 269 | 26.7 |
| | Average | 424 | 42.0 |
| | Small | 316 | 31.3 |
| Had diarrhoea* | Yes | 160 | 15.9 |
| | No | 849 | 84.1 |
| Had cough* | Yes | 137 | 13.6 |
| | No | 872 | 86.4 |

*Diarrhoea and cough were assessed for two weeks preceding the survey.

**Table 4. Community-level variables in the emerging regions of Ethiopia, EDHS 2016 (n = 1009).**

| Variables | Category | Frequency (n) | Percent (%) |
|---|---|---|---|
| Residence | Urban | 163 | 16.1 |
| | Rural | 846 | 83.9 |
| Region | Afar | 254 | 25.2 |
| | Somali | 346 | 34.3 |
| | Benishangul | 224 | 22.2 |
| | Gambela | 185 | 18.3 |
| Community level wealth quantile | Poorest | 637 | 63.1 |
| | Poorer | 162 | 16.1 |
| | Middle | 94 | 9.3 |
| | Richer | 74 | 7.3 |
| | Richest | 42 | 4.2 |
| Community level media exposure | Yes | 50 | 5.0 |
| | No | 959 | 95.0 |

index, place of delivery, PNC visit, desire more child, child currently breastfeed, currently pregnant mother, diarrhoea and cough in the last two weeks, community level poverty, and community level media exposure were not significant with the MN intake among children aged 6–23 months.

Accordingly, the odds of recommended MN intake among children whose mothers/caregivers with an agricultural occupation were 2.2 times higher than those whose mothers/

**Table 5. Micronutrient intake status among children aged 6–23 months in the emerging regions of Ethiopia, 2016 (n = 1009).**

| Food groups and supplementations | Contains/measurements | Received | |
|---|---|---|---|
| | | n | % (95% CI) |
| Consumed foods rich in VA within 24 hours | Eggs | 85 | 8.4 (6.8–10.3) |
| | Meat (beef, pork, lamb, chicken, etc) | 52 | 5.2 (3.9–6.7) |
| | Pumpkin, carrots, and squash | 111 | 11.0 (9.2–13.0) |
| | Any dark green leafy vegetables | 91 | 9.0 (7.4–10.9) |
| | Mangoes, papayas, and others with VA fruits | 133 | 13.2 (11.2–15.4) |
| | Liver, heart, and other organs | 32 | 3.2 (2.2–4.4) |
| | Fish or shellfish. | 45 | 4.5 (3.3–5.9) |
| Overall VA rich foods consumptions | | 280 | 27.7 (25.0–30.5) |
| Consumed foods rich in iron at any time in 24 hours | Eggs | 85 | 8.4 (6.8–10.3) |
| | Meat (beef, pork, lamb, chicken) | 52 | 5.2 (3.9–6.7) |
| | Liver, heart, and other organs | 32 | 3.2 (2.2–4.4) |
| | Fish or shellfish. | 45 | 4.5 (3.3–5.9) |
| Overall iron rich food consumption | | 157 | 15.6 (13.3–17.8) |
| Multiple micronutrient powder within seven days | | 76 | 7.5 (6.1–9.3) |
| Iron supplements within seven days | | 60 | 6.0 (4.6–7.6) |
| VAS within six months | | 476 | 47.2 (44.1–50.3) |
| Deworming medication in the six months (n = 547) | | 46 | 8.4 (7.7–8.9) |
| Overall, received at least one of the recommended MNs | | 633 | 62.7 (59.7–65.7) |

MNs: Micronutrients.

VA: Vitamin A.

VAS: Vitamin A Supplements.

**Table 6. Results from a random intercept model (a measure of variation) for MN intake among children aged 6–23 months at cluster level by multilevel logistic regression analysis, EDHS 2016.**

| Measure of variations | Model 0 (null model) | Model 1 | Model 2 | Model 3 (full model) |
|---|---|---|---|---|
| Variance | 3.35 | 1.49 | 1.61 | 1.43 |
| Explained variation (PCV) (%) | Ref. | 55 | 52 | 57 |
| Model fitness | | | | |
| Deviance (-2*log likelihood) | 1271.9 | 1154.7 | 1195.0 | 1135.4 |
| AIC | 1275.9 | 1200.4 | 1214.5 | 1193.8 |

AIC: Akaike's Information Criterion.

ICC: Intra-class Correlation Coefficient.

PCV: Proportional Change in Variance.

Model 0: Without independent variables (null model).

Model 1: Only individual-level variables.

Model 2: Only community-level variables.

Model 3: Individual and community-level variables (full model).

caregivers with no work (AOR: 2.2, 95% CI: 1.3–3.8). Children born from mothers who had ANC visits for their recent pregnancy were had 1.9 times more odds to receive any one of the six recommended MNs than those who had no ANC visits (AOR: 1.9, 95% CI:1.4–2.8). Those children aged 13 to 23 months were had 1.7 times more odds to receive the recommended MN compared to those aged 6 to 12 months (AOR: 1.7, 95% CI: 1.2–2.4). Those children who reside in the rural communities were 60% lower to receive any MNs than urban residents (AOR: 0.4, 95% CI: 0.1–0.9). The odds of taking any one of the MNs among children who live in the Benishangul and Gambella region were 2.5 (AOR: 2.5, 95% CI: 1.3–4.9) and 1.9 (AOR: 1.9, 95% CI: 1.0–3.4) times higher than those children who live in the Afar region, respectively (Table 7).

## Discussion

The study showed that 37.3% of children aged 6–23 months had not received any to the recommended MNs sources in the emerging regions of Ethiopia. After adjusting for individual and community level factors, women's occupational status, the child's age, antenatal visits for current pregnancy, residence and region were significantly associated with the MN intake status among children aged 6–23 months. In this study, 28.0% and 15.6% of children had consumed foods rich in VA and iron, respectively. The EDHS 2016 showed that consumption of foods rich in VA and iron was 38.0% and 22.0%; correspondingly, the lowest intake was observed in Afar [23], comparable with the current finding. Almost half of the children (47.2%) got VAS and as few as 6.0% of them got iron supplements. The previous EDHS (2011) finding showed that VAS in the four regions was 43.2%, which is lower than our findings [35].

This study identified that MN intake among children from mothers who had no formal/paid jobs was lower than children whose mothers had work. Mothers who work in agriculture might have better access to diversified agricultural and animal products rich sources of MNs. Moreover, participating in work may expose mothers to peers and friends who can serve as sources of information related to MN intake and its benefits. This study also showed that agrarian dominants were more likely to consume diversified food, which can be used as a proxy for adequate MN density of foods [41]. Previous studies in Ethiopia and Nigeria are also consistent with this study [35,42].

**Table 7. Multilevel mixed effect logistic regression analysis of factors associated with MN intake status among children aged 6–23 months in the emerging regions of Ethiopia, EDHS 2016 (n = 1009).**

| Variables | Received at least one of the recommended MNs | | COR (95%CI) | Model 1 AOR (95% CI) | Model 2 AOR (95%CI) | Model 3 AOR (95%CI) |
|---|---|---|---|---|---|---|
| | Yes n (%) | No n (%) | | | | |
| Individual-level characteristics | | | | | | |
| Mothers' occupation | | | | | | |
| No work | 371 (55.3) | 300 (44.7) | 1 | 1 | | 1 |
| Professional | 57 (71.3) | 23 (28.7) | 1.8 (0.9–3.2) | 1.4 (0.7–2.5) | | 1.4 (0.8–2.5) |
| Agricultural | 154 (81.5) | 35 (18.5) | 3.3 (2.1–5.4) | 3.0 (1.9–4.9) | | 2.2 (1.3–3.8) * |
| Others | 51 (73.9) | 18 (26.1) | 2.0 (1.1–3.9) | 1.5 (0.8–2.9) | | 1.3 (0.7–2.6) |
| ANC visit | | | | | | |
| No | 214 (48.6) | 226 (51.4) | 1 | 1 | | 1 |
| Yes | 419 (73.6) | 150 (26.3) | 2.8 (2.0–3.8) | 2.0 (1.4–2.8) | | 1.9 (1.4–2.8) * |
| Age of child in months | | | | | | |
| 6–12 | 259 (56.1) | 203 (43.9) | 1 | 1 | | 1 |
| 13–23 | 374 (68.4) | 173 (31.6) | 1.7 (1.3–2.4) | 1.8 (1.3–2.5) | | 1.7 (1.2–2.4) * |
| Community-level characteristics | | | | | | |
| Residence | | | | | | |
| Urban | 126 (77.3) | 37 (22.7) | 1 | | 1 | 1 |
| Rural | 507 (60.0) | 339 (40.0) | 0.4 (0.2–0.8) | | 0.4 (0.2–0.7) | 0.4 (0.1–0.9) * |
| Region | | | | | | |
| Afar | 127 (50.0) | 127 (50.0) | 1 | | 1 | 1 |
| Somali | 182 (52.6) | 164 (47.4) | 1.1 (0.7–1.8) | | 1.0 (0.6–1.6) | 1.1 (0.7–1.76) |
| Benishangul | 187 (83.4) | 37 (16.6) | 6.4 (3.5–11.7) | | 5.3 (2.9–9.7) | 2.5 (1.3–4.9) * |
| Gambella | 137 (74.0) | 48 (26.0) | 3.81 (2.1–6.9) | | 2.9 (1.6–5.0) | 1.9 (1.0–3.4) * |

*Statistically significant at p-value <0.05 at model 3.

ANC: Antenatal Care.

AOR: Adjusted Odds Ratio.

COR: Crude Odds Ratio.

The odds of MN intake for children aged 13–23 were higher than those aged between 6 and 12 months. This could be explained by poor complementary feeding practices that should be introduced at six months of age, especially in the rural population and emerging regions. Also, older age groups could have better dietary diversity as they can eat family meals for themselves. In EDHS 2016, children above 12 months old were more likely to obtain diversified food [41,43]. The late introduction of complementary feeding might have resulted in consuming a limited variety of food, such as only milk or cereal products. Moreover, mothers' perceptions and traditional beliefs might contribute to low consumption of diversified food in those children (6–12 months).

In this study, higher odds of MN intake were observed among children whose mothers had ANC follow-ups compared to those whose mothers did not have ANC follow-ups. This finding was in line with those of previous studies conducted in Ethiopia [35]. The possible explanation might be that mothers who had ANC follow-up may have a chance to get information, education, knowledge, and counselling services from the health professionals. Caregivers may have learned or acquired knowledge of iron supplements during their ANC follow-up. Another explanation might be that mothers with follow-up live nearer to health facilities have more

time/money available to attend ANC. Moreover, a systematic review and meta-analysis of dietary diversity feeding practice done in Ethiopia suggests that children whose mothers have ANC follow-up have a higher probability than their counterparts to eat diversified food [44].

The odds of MN intake among children who reside in rural communities were lower compared to their counterparts. This is supported by a systematic study in Ethiopia, which reported that urban residents had higher odds of MN intake than rural residents [44]. However, a few studies' findings [45–47] contradict the current study. The potential explanation might be food fortification and supplementation focused more on rural than urban through community-based maternal and child health outreach programs.

Our finding showed that MN intake among children living in Benishangul and Gambella regions was higher than those who live in the Afar region. This can be explained by the fact that, compared to the two regions, the Afar and Somalia regions' economic activities are mostly dominated by cattle breeding and pastoral lifestyles, and agriculture is common in Benishangul and Gambella. Besides, since the latter two regions have dense forests and water reservoirs, caregivers could get wild fruit and fish, which are good sources of MNs. Previous studies showed that VA rich foods were scarce in the pastoral community, and meat and egg consumption were low [48]. Natural forests and semi-natural forests were positively associated with many nutritionally important food groups [49]. A study from the recent EDHS (2016) showed that the agrarian community children were more likely to consume diversified food than the pastoral community.

The study's main strengths are its representativeness, large sample size, and the availability of individual and community-level factors. This study used a multilevel-modelling technique to identify a more valid result that takes the survey data's hierarchical nature into account. Furthermore, the DHS methodology allows for comparison with other settings. The mothers might have experienced recall bias, particularly regarding VAS and deworming for their child in the last six months before the survey, for instance.

## Conclusions

The overall intake of MNs in this study was below the national recommendation. Mothers' occupation, age of a child, recent ANC, residence, and region were significantly associated with the MN intake status. Improving ANC, promoting affordable and available MN-rich foods through improved/adaptive agricultural practices, deworming, MNPs, Iron and VAS are essential for increasing MN intake among children in Ethiopia.

## Acknowledgments

We are very thankful to MEASURE DHS for permission to use the EDHS 2016 survey data sets.

## Author Contributions

**Conceptualization:** Tsegaye Gebremedhin.

**Data curation:** Tsegaye Gebremedhin.

**Formal analysis:** Tsegaye Gebremedhin, Andualem Yalew Aschalew, Chalie Tadie Tsehay, Endalkachew Dellie, Asmamaw Atnafu.

**Methodology:** Tsegaye Gebremedhin, Andualem Yalew Aschalew, Chalie Tadie Tsehay, Endalkachew Dellie, Asmamaw Atnafu.

**Writing – original draft:** Tsegaye Gebremedhin, Andualem Yalew Aschalew, Chalie Tadie Tsehay, Endalkachew Dellie, Asmamaw Atnafu.

**Writing – review & editing:** Tsegaye Gebremedhin, Andualem Yalew Aschalew, Chalie Tadie Tsehay, Endalkachew Dellie, Asmamaw Atnafu.

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
