## [Decision Letter · Decision Letter 0]

15 Jun 2020

PONE-D-20-13718

Antenatal care visit increases micronutrient intake among children aged 6-23 months in the emerging regions of Ethiopia: a multilevel analysis of the 2016 Ethiopian demographic and health survey

PLOS ONE

Dear Dr. Tsegaye Gebremedhin,

Thank you for submitting your manuscript to PLOS ONE. After careful consideration, we feel that it has merit but does not fully meet PLOS ONE’s publication criteria as it currently stands. Therefore, we invite you to submit a revised version of the manuscript that addresses the points raised during the review process.

Please pay careful attention to the comments made by both reviewersGeneral improvements in English are recommended before re-sbvmission

We look forward to receiving your revised manuscript.

Kind regards,

Mary Hamer Hodges

Academic Editor

PLOS ONE

Journal Requirements:

2. Please modify the title to ensure that it is meeting PLOS’ guidelines (https://journals.plos.org/plosone/s/submission-guidelines#loc-title). In particular, the title should be "specific, descriptive, concise, and comprehensible to readers outside the field" and in this case we have concerns that the title contains a causal statement not completely supported by the study.

Additional Editor Comments (if provided):

Reviewers' comments:

Reviewer's Responses to Questions

**Comments to the Author**

1. Is the manuscript technically sound, and do the data support the conclusions?

Reviewer #1: Partly

Reviewer #2: Partly

2. Has the statistical analysis been performed appropriately and rigorously? 

Reviewer #1: Yes

Reviewer #2: Yes

3. Have the authors made all data underlying the findings in their manuscript fully available?

Reviewer #1: Yes

Reviewer #2: Yes

4. Is the manuscript presented in an intelligible fashion and written in standard English?

Reviewer #1: Yes

Reviewer #2: No

5. Review Comments to the Author

Reviewer #1: 1. I do not find there to be a strong rationale in the paper for why only data from four regions is analysed, when (presumably) the EDHS has data available from other regions as well.

2. Dependent variable. The individual questions are well explained but I do not fully understand how you combined these into the outcome variable. In the paper you refer both to micronutrient "intake status", which could be categorical, but also "level of intake" which sounds continuous; however my understanding is your outcome variable is binary, in which case "minimum intake" or similar would be most appropriate. Furthermore, was a positive answer to any of the six questions sufficient for the outcome to be coded as a "yes"? On a conceptual level, I find it difficult to understand why this makes sense if the child has received deworming medication but answers to the five other questions are negative.

3. Sampling procedures. Please use the term "community" in this section, since this is used frequently elsewhere in the paper. Is EA the community level?

4. Rows 123-125: Please clarify whether 1009 was the total number of eligible respondents available in the EDHS (from the four regions), or whether you have made some selection.

5. The description of multilevel modelling is very nice and while I am not overly familiar I believe there is a mistake on row 200, where the reference should be to (beta0+Uj) and beta2.

6. The large number of variables describing obstetric history is surprising, given the outcome variable. No rationale is provided in the paper either, and very few end up being included in the final model.

7. Discussion. You mention there is a national target, however, I cannot find a mention of what the target actual is/was.

8. Strengths and weaknesses. Given the focus on four regions, I am not convinced by your argument that the results are nationally representative, nor that the use of standardised DHS questions makes findings generalisable to other settings. Instead, it would be relevant to discuss strengths and weaknesses of using EDHS as a data source, compared to alternatives e.g. cross-sectional survey focused on nutrition, to answer the research question.

9. Conclusion. You recommend specifically continued investment in scaling up vitamin A supplementation, but I cannot find any support in your results to suggest that this would be more important a focus than say iron supplements.

10. Data availability. This needs to be revised to refer readers to Measure DHS.

11. There are minor spelling mistakes namely "ricked" (row 60), "antennal" (row 305), "vibrations" (Table 5). In references, WHO shows up as Organization WH.

Reviewer #2: • Line 166-167: Can you clearly state whether Vitamin A uptake and ANC visits was based on mother’s recall, some form of Health documents (like child health card) or both. What about distance to the health facility? This is an ideal variable when it comes to mother’s visits to the clinics. I advise you adequately described, analyzed and discussed if the data exist

• Table 4: Please include confidence intervals for the coverages

• The sample size is not clear. Can you please clarify whether 1009 was the total number of eligible respondents available from the four regions? Also you have used same denominators for mothers and children sample characteristics. It is possible that a mother was interviewed for more than one child. Can you please clarify here

• Discussion: You have only stated the national target but have not provided or given a reference for that.

• There are a few typos and grammatical error. Please get a native English speaker to review and help correct some of the minor mistakes.

• Strengths and weaknesses: I find it problematic that you have not used the 6 months recall for Vitamin A. I assume the EDHS used the 6 months recall and so I do not think your argument that the results can be compared to national estimates is right especially given the fact that there is some recall bias here.

• Please provide p values for comparism where there are significant differences.

• Conclusion: What is the national vitamin A coverage and how does it compare to the findings here? Also worth citing the national Vitamin A deficiency status from a most recent micronutrient survey. Then you need to make this comparism clear before you can recommend continued investment in scaling up vitamin A supplementation. I also think promoting Vitamin A rich foods is a good strategy worth promoting but that will depend on the status of vitamin A in the country.

6. PLOS authors have the option to publish the peer review history of their article (what does this mean?). If published, this will include your full peer review and any attached files.

Reviewer #1: Yes: Anni-Maria Pulkki-Brännström

Reviewer #2: Yes: Habib Issa Kamara

---

## [Author Response · Author response to Decision Letter 0]

1 Aug 2020

Dear Editor,

Greetings!

Firstly, we would like to appreciate and thank the academic editor and reviewers for investing their time and energy to review and make comments. It is with great pleasure to receive the invaluable and constructive comments which improves our manuscript. We accepted and tried to incorporate all of the comments provided. Moreover, the manuscript has been revised by English language expert and grammar and spellings have been improved throughout the manuscript, and we made rewording and rephrasing some parts of the paragraphs of the paper accordingly. The responses to the editor and reviewers' comments are provided here below; please see the responses.

1-When submitting your revision, we need you to address these additional requirements:

Please ensure that your manuscript meets PLOS ONE's style requirements, including those for file naming. The PLOS ONE style templates can be found at http://www.plosone.org/attachments/PLOSOne_formatting_sample_main_body.pdf and http://www.plosone.org/attachments/PLOSOne_formatting_sample_title_authors_affiliations.pdf

Authors' response: Thank you so much for your comments. The comment is accepted and corrected in the style of the PLOS ONE journal. Please refer to the clean version of the revised manuscript.

2- Please modify the title to ensure that it is meeting PLOS’ guidelines (https://journals.plos.org/plosone/s/submission-guidelines#loc-title) . In particular, the title should be "specific, descriptive, concise, and comprehensible to readers outside the field" and in this case we have concerns that the title contains a causal statement not completely supported by the study

Authors' response: Again, thank you very much for your very kindly and careful review. We had gone throughout the linked document and we have modified the title “Micronutrient intake status and associated factors among children aged 6-23 months in the emerging regions of Ethiopia: a multilevel analysis of the 2016 Ethiopian demographic and health survey” to be more specific, descriptive, concise and comprehensive with the findings. Please see the first page of the clean version of the revised manuscript.

Reviewer #1: 

1. I do not find there to be a strong rationale in the paper for why only data from four regions is analysed, when (presumably) the EDHS has data available from other regions as well.

Authors’ response: Dear reviewer, thank you for your constructive comments. We revised the introduction and set out the rationale of the study as per the comments provided; please see the clean version of the revised manuscript on page 5, lines 72-81.

2. Dependent variable. The individual questions are well explained but I do not fully understand how you combined these into the outcome variable. In the paper you refer both to micronutrient "intake status", which could be categorical, but also "level of intake" which sounds continuous; however, my understanding is your outcome variable is binary, in which case "minimum intake" or similar would be most appropriate. Furthermore, was a positive answer to any of the six questions sufficient for the outcome to be coded as a "yes"? On a conceptual level, I find it difficult to understand why this makes sense if the child has received deworming medication but answers to the five other questions are negative.

Authors’ response: Dear reviewer, thank you very much for your comments and insights. The dependent variable of the study was categorical; It was dichotomized in yes and no category. when one chid got at least one of the minimum recommended micronutrients, they were considered us “got the recommended micronutrient”; otherwise it was considered as that “the recommended micronutrient is not received”. We used the classification according to the national micronutrient recommendations, and if the child received at least one of the minimum recommended micronutrients, the national recommendation stated that they were considered “received”; otherwise “not received”. Such recommendation might not give confidence to say the micronutrient intake status is sufficient, so, this might be one of the limitations of the study mentioned on the strength and limitation section (page 26, lines 404-406).

Furthermore, for the deworming medication, it was not considered the medication prescribed for treatment of disease/illness; rather the national micronutrient recommendation has included it, for any heathy child, it should be supplement alongside with the vitamin A and others. So, we have included the deworming medication which was provided as a supplement rather than for those prescribed for the treatment of illness/disease. Please refer to the variables' measurement section in the clean version of the revised manuscript on pages 8-11, lines 132-183.

3. Sampling procedures. Please use the term "community" in this section, since this is used frequently elsewhere in the paper. Is EA the community level?

Authors’ response: Dear reviewer, thank you for your concerns. In the EDHS, enumeration areas were the clusters with a specified household numbers; thus, instead of clusters we used as EA. Furthermore, EA are not the community level rather they are clusters created in the Ethiopian demographic and health survey for sampling purposes. So, we kindly invite you to see the sampling procedures as per those perspectives, on pages 6-7, lines 95-116 in the revised manuscript.

4. Rows 123-125: Please clarify whether 1009 was the total number of eligible respondents available in the EDHS (from the four regions), or whether you have made some selection.

Authors response: Dear reviewer, we are thankful for your positive comments to improve the manuscript. In the first step, a total of 1130 participants from the four regions were available as a source. Then, using a selection criterion; alive, first index to birth history and live with their mothers’/caregivers, we have included a total of 1009 eligible children with their mothers/caregivers in the final analysis. we put this in figure 1; please see the clean version of the revised manuscript on page 7, lines 117-124.

5. The description of multilevel modelling is very nice and while I am not overly familiar, I believe there is a mistake on row 200, where the reference should be to (beta0+Uj) and beta2.

Authors response: Dear reviewer, we are very much thankful for your observation and comments. We have addressed the issue; please see the clean version of the revised manuscript on page 11, line 203.

6. The large number of variables describing obstetric history is surprising, given the outcome variable. No rationale is provided in the paper either, and very few end up being included in the final model.

Authors response: Dear reviewer, thank you for your critical insights and suggestions. In our assumptions, the micronutrient intake status of children can be affected by the maternal obstetric history as we tried to show in the introduction section (page 5, lines 69-71). Moreover, we have revised the obstetric history in the result section and removed few irrelevant variables form the descriptive; please see the clean version of the revised manuscript on page 13, lines 237-243.

7. Discussion. You mention there is a national target, however, I cannot find a mention of what the target actual is/was.

Authors response: Thank you for your comments. We mentioned the national targets; please see the clean version on page 23, line 345 of the revised manuscript. Besides, we included that in the introduction section (page 5, lines 65-66).

8. Strengths and weaknesses. Given the focus on four regions, I am not convinced by your argument that the results are nationally representative, nor that the use of standardised DHS questions makes findings generalisable to other settings. Instead, it would be relevant to discuss strengths and weaknesses of using EDHS as a data source, compared to alternatives e.g. cross-sectional survey focused on nutrition, to answer the research question.

Authors response: Dear reviewer, thank you for your detailed evaluation. We have addressed the issue; please see the clean version of the manuscript on page 26, lines 396-410.

9. Conclusion. You recommend specifically continued investment in scaling up vitamin A supplementation, but I cannot find any support in your results to suggest that this would be more important a focus than say iron supplements.

Authors response: Dear reviewer, thank you so much. We have revised the conclusion section and the problem has been addressed; please see the clean version of the revised manuscript on pages 26-27, lines 412-421.

10. Data availability. This needs to be revised to refer readers to Measure DHS.

Authors response: Thank you for your comments. We have addressed the issue.

11. There are minor spelling mistakes namely "ricked" (row 60), "antennal" (row 305), "vibrations" (Table 5). In references, WHO shows up as Organization WH.

Authors response: Dear reviewer, thank you for your insights, we have addressed the issue throughout the manuscript please see the clean version of the revised manuscript.

Reviewer #2: 

1. Line 166-167: Can you clearly state whether Vitamin A uptake and ANC visits was based on mother’s recall, some form of Health documents (like child health card) or both. What about distance to the health facility? This is an ideal variable when it comes to mother’s visits to the clinics. I advise you adequately described, analyzed and discussed if the data exist

Authors response: Dear reviewer, thank you very much for your constructive comments. We have addressed the issue; please see the measurement of variables section, particularly on page 9-10, lines 159-183.

2. Table 4: Please include confidence intervals for the coverages

Authors response: Dear reviewer, thank you for your comments. We have included the confidence intervals; please see the clean version of the revised manuscript on page 15-16, lines 260-271.

3. The sample size is not clear. Can you please clarify whether 1009 was the total number of eligible respondents available from the four regions? Also, you have used same denominators for mothers and children sample characteristics. It is possible that a mother was interviewed for more than one child. Can you please clarify here

Authors response: Again, thank you very much for your very kindly and careful review. We described the EDHS sample selection in text, and presented it in figure as well. Please see the clean version of the revised manuscript on page 7, lines 117-124, and also presented in figure 1.

4. Discussion: You have only stated the national target but have not provided or given a reference for that.

Authors response: Dear reviewer, thank you for your insights. We have addressed the issue; please see the clean version of the revised manuscript on page 23, line 345.

5• There are a few typos and grammatical error. Please get a native English speaker to review and help correct some of the minor mistakes.

Authors response: Dear reviewer, thank you so much for your comments and suggestions. The manuscript has been revised by English language expert and grammar and spellings have been improved throughout the manuscript, and we made rewording and rephrasing some parts of the paragraphs of the paper accordingly; please see the clean version of the revised manuscript once again.

6. Strengths and weaknesses: I find it problematic that you have not used the 6 months recall for Vitamin A. I assume the EDHS used the 6 months recall and so I do not think your argument that the results can be compared to national estimates is right especially given the fact that there is some recall bias here.

Authors response: Dear reviewer, thank you for your valuable comments. We have revised the strength and weaknesses of our study; please see the clean version of the revised manuscript on page 26, lines 396-410.

7. Please provide p values for comparism where there are significant differences.

Authors response: Dear reviewer, thank you for your comments and the comment is admitted. We have used the confidence interval and also the p values for the comparison, but to be consistent in writing/editing of the manuscript we did not mentioned it. So, kindly request you to consider it.

8. Conclusion: What is the national vitamin A coverage and how does it compare to the findings here? Also, worth citing the national Vitamin A deficiency status from a most recent micronutrient survey. Then you need to make this comparism clear before you can recommend continued investment in scaling up vitamin A supplementation. I also think promoting Vitamin A rich foods is a good strategy worth promoting but that will depend on the status of vitamin A in the country.

Authors response: Dear reviewer, we are very thankful for your comments and suggestions. We have addressed it as per the comments; please see the clean version of the revised manuscript on pages 26-27, lines 412-421.

---

## [Decision Letter · Decision Letter 1]

26 Aug 2020

PONE-D-20-13718R1

Micronutrient intake status and associated factors among children aged 6-23 months in the emerging regions of Ethiopia: a multilevel analysis of the 2016 Ethiopian demographic and health survey

PLOS ONE

Dear Tsegaye Gebremedhin,

Thank you for submitting your manuscript to PLOS ONE. After careful consideration, we feel that it has merit but does not fully meet PLOS ONE’s publication criteria as it currently stands. Therefore, we invite you to submit a revised version of the manuscript that addresses the points raised during the review process.

We look forward to receiving your revised manuscript.

Kind regards,

Mary Hamer Hodges, MBBS MRCP DSc

Academic Editor

PLOS ONE

Additional Editor Comments (if provided):

I should double check that you are entitled to provide the dataset as supporting information file by contacting Measure DHS.

Reviewers' comments:

Reviewer's Responses to Questions

**Comments to the Author**

1. If the authors have adequately addressed your comments raised in a previous round of review and you feel that this manuscript is now acceptable for publication, you may indicate that here to bypass the “Comments to the Author” section, enter your conflict of interest statement in the “Confidential to Editor” section, and submit your "Accept" recommendation.

Reviewer #1: (No Response)

Reviewer #2: All comments have been addressed

2. Is the manuscript technically sound, and do the data support the conclusions?

Reviewer #1: Yes

Reviewer #2: Yes

3. Has the statistical analysis been performed appropriately and rigorously? 

Reviewer #1: Yes

Reviewer #2: Yes

4. Have the authors made all data underlying the findings in their manuscript fully available?

Reviewer #1: Yes

Reviewer #2: Yes

5. Is the manuscript presented in an intelligible fashion and written in standard English?

Reviewer #1: No

Reviewer #2: Yes

6. Review Comments to the Author

Reviewer #1: 1) I am surprised that a data file is provided as supporting information because as the authors correctly report, the data can be very easily accessed through contacting MeasureDHS. I would recommend the authors double check that they are entitled to provide the dataset as supporting information file.

Otherwise, I only have two minor concerns that relate to ambiguities remaining after the authors' otherwise satisfactory responses (to my previous comments nr 2 and 4):

2) References to “received the recommended micronutrients” can be misunderstood to mean adequate intake. Please consistently refer to “received at least one of the recommended micronutrients” to ensure no reader is mislead about your outcome variable. For example: in the results section of the Abstract, in the column heading in Table 6, and the first sentence of the Discussion.

3) Regarding the exclusion criteria, please rephrase “index to birth history” in Figure 1 and on row 120, because the meaning is difficult to understand.

Reviewer #2: In the results section you stated children born of mothers who had ANC visits for their recent pregnancy were 1.95 times more likely to receive micronutrients. This may be related to knowledge gained from health talks during ANC visits. It would be prudent to proffer reasons for these differences.

Also proffer reasons in the discussion section for why more mother tend to do complementary feeding for children ages 13-23 compared to 6-12months.

I think it would be worth knowing why more mother preferred to give birth at compared to health facility. Is this because of distance to health facility, costs or they trust the traditional birth attendance more than they do health workers?

7. PLOS authors have the option to publish the peer review history of their article (what does this mean?). If published, this will include your full peer review and any attached files.

Reviewer #1: **Yes: **A-M Pulkki-Brannstrom

Reviewer #2: No

---

## [Author Response · Author response to Decision Letter 1]

29 Sep 2020

Response to reviewers’

To: PLOS ONE Journal Editorial Office

Manuscript title: “Micronutrient intake status and associated factors among children aged 6-23 months in the emerging regions of Ethiopia: a multilevel analysis of the 2016 Ethiopian demographic and health survey” [Manuscript ID: PONE-D-20-13718].

Subject: Submission of a revised manuscript for publication

Dear Editor,

Greetings!

We appreciate and acknowledge the academic editor and reviewers for investing their time and energy to review and make comments on our manuscript once again. It is with great pleasure to receive the invaluable and constructive comments for our manuscript. 

We accepted and tried to incorporate all of the comments provided. Moreover, the manuscript has been revised by English language expert and grammar and spellings have been improved throughout the manuscript. Thus, the comments are attached here below with their point-by-point responses. In addition, the detailed changes made are highlighted in the “revised manuscript with track changes” to easily identify the changes/improvements and also the clean copy of the revised manuscript is prepared. 

Finally, we kindly request you to review our revised manuscript.

 

RESPONSE TO EDITOR’S COMMENTS

Academic editor (Mary Hamer Hodges, MBBS MRCP DSc)

#1. I should double check that you are entitled to provide the dataset as supporting information file by contacting Measure DHS.

Authors’ response: Dear editor, thank you for your important comment. We have double checked the authorization letter, and unfortunately, we were prohibited to share the data set, “The data must not be passed on to other researchers without the written consent of DHS”. So, we would like to say sorry for the previous data set attachment as a supplementary information and we have removed the data set from the manuscript tracking system. Finally, we kindly request the journal office to remove the data set from the public repository if it’s deposited. 

Response to Reviewers 

Reviewer #1 (Anni-Maria Pulkki-Brannstrom):

1. I am surprised that a data file is provided as supporting information because as the authors correctly report, the data can be very easily accessed through contacting Measure DHS. I would recommend the authors double check that they are entitled to provide the dataset as supporting information file.

Authors’ response: Dear reviewer, we are very much thankful for your critical insights. We have checked the authorization letter; unfortunately, we were prohibited to share the data set, “The data must not be passed on to other researchers without the written consent of DHS”. So, we have removed the data from the supporting information and we hope you will definitely understand for the mistaken done in our previous submission regarding to the data file. Finally, we kindly request the journal editorial office to remove the data set from the supplementary files.

2. References to “received the recommended micronutrients” can be misunderstood to mean adequate intake. Please consistently refer to “received at least one of the recommended micronutrients” to ensure no reader is misled about your outcome variable. For example: in the results section of the Abstract, in the column heading in Table 6, and the first sentence of the discussion.

Authors’ response: Dear reviewer, thank you for your comment. We have amended the term of outcome variable “received the recommended micronutrients” in to “received at least one of the recommended micronutrients” in the results section of the Abstract, in the column heading in Table 6, and the first sentence of the discussion. Kindly see the clean version of the revised manuscript on page 2 lines 24-25, Table 6 column heading on page 20 and in the discussion page 23 lines 322-323.

3. Regarding the exclusion criteria, please rephrase “index to birth history” in Figure 1 and on row 120, because the meaning is difficult to understand.

Authors’ response: Dear reviewer, thank you so much for your comments. We have addressed the issue as per the comments, kindly see the clean version of the revised manuscript on page 7, lines 119-120.

Reviewer #2:

1. In the results section you stated children born of mothers who had ANC visits for their recent pregnancy were 1.95 times more likely to receive micronutrients. This may be related to knowledge gained from health talks during ANC visits. It would be prudent to proffer reasons for these differences.

Authors’ response: Dear reviewer, thank you very much for your comment. We have mentioned that information and knowledge gained from health talks during ANC visits could be the possible explanations of micronutrient intake status difference. Kindly see the clean version of the revised manuscript on page 25, lines 373-376.

2. Also proffer reasons in the discussion section for why more mother tend to do complementary feeding for children ages 13-23 compared to 6-12months.

Authors’ response: Dear reviewer, thank you for your comments; we have included the possible explanation. We tried to discuss the difference in feeding practice by stating the contributor factors of low feeding among children 6-12 months. Please see the clean version of the revised manuscript on page 24, lines 363-366.

3. I think it would be worth knowing why more mother preferred to give birth at compared to health facility. Is this because of distance to health facility, costs or they trust the traditional birth attendance more than they do health workers?

Authors’ response: Dear reviewer, thank you very much for your insight. A plenty of studies identified that distance to health facility, mothers’ low awareness about institutional delivery and others cultural and social factors were the high contributor for having a significant number of home delivery in Ethiopia. These factors were contributed not only for low institutional delivery services utilization and or coverage, but also for other maternal and neonatal health services uptake like postnatal services.

---

## [Decision Letter · Decision Letter 2]

20 Nov 2020

PONE-D-20-13718R2

Micronutrient intake status and associated factors among children aged 6-23 months in the emerging regions of Ethiopia: a multilevel analysis of the 2016 Ethiopian demographic and health survey

PLOS ONE

Dear Dr. Tsegaye Gebremedhin,

Thank you for submitting your manuscript to PLOS ONE. After careful consideration, we feel that it has merit but does not fully meet PLOS ONE’s publication criteria as it currently stands. Therefore, we invite you to submit a revised version of the manuscript that addresses the points raised during the review process.

The definition of to recommended MN interventions needs to be clearly introducedWhy have you included de-worming?The discussion needs to be better oragnized to address first the findings and then comparisons with other manuscripts on MNs.

We look forward to receiving your revised manuscript.

Kind regards,

Mary Hamer Hodges, MBBS MRCP DSc

Academic Editor

PLOS ONE

Reviewers' comments:

Reviewer's Responses to Questions

**Comments to the Author**

1. If the authors have adequately addressed your comments raised in a previous round of review and you feel that this manuscript is now acceptable for publication, you may indicate that here to bypass the “Comments to the Author” section, enter your conflict of interest statement in the “Confidential to Editor” section, and submit your "Accept" recommendation.

Reviewer #1: All comments have been addressed

Reviewer #2: All comments have been addressed

Reviewer #3: All comments have been addressed

2. Is the manuscript technically sound, and do the data support the conclusions?

Reviewer #1: (No Response)

Reviewer #2: Yes

Reviewer #3: Yes

3. Has the statistical analysis been performed appropriately and rigorously? 

Reviewer #1: (No Response)

Reviewer #2: Yes

Reviewer #3: Yes

4. Have the authors made all data underlying the findings in their manuscript fully available?

Reviewer #1: (No Response)

Reviewer #2: No

Reviewer #3: Yes

5. Is the manuscript presented in an intelligible fashion and written in standard English?

Reviewer #1: (No Response)

Reviewer #2: No

Reviewer #3: Yes

6. Review Comments to the Author

Reviewer #1: (No Response)

Reviewer #2: My previous comments have all been addressed. I think there are a few areas with grammatical and typo errors especially in the discussion which needs attention. I advise the authors to find another native/fluent English language speaker to review this thoroughly.

In the discussion, please provide adequate references where comparism is made to other studies.

Reviewer #3: Comments

L 2: It is unclear what “emerging” means in this context. Please provide an explanation when this word is first introduced.

ABSTRACT

L13: Scarcity of MN is due to poor diets rather than poor health care access, health care access helps mitigate the impact of these poor diets but are not the fundamental cause.

L15: children 6-23 months include “and their associated factors”. Which MN? it is essential you specify what you are measuring is it Vitamin A supplementation or something else: MNPs, 24 dietary recall?

L 25: this is too generic, need to specify which ‘recommended micronutrient supplements’. If more than just VAS, what else?

L 30: Again, still unclear whether you are describing VAS or MNPs or 24 hour dietary recall.

L31: i suggest "study population in this area", rather than "study area"

L33: i suggest putting a full stop (.) after ‘community level’

L35: This final sentence is a recommendation, please conclude your findings rather than just saying they were low in the previous sentence.

INTRODUCTION

L45: I suggest “micronutrients are”

L50: I suggest “growth” instead of grow

L55: please define Minimal Diversified Diets (MDD)

L66: You need to explain more about VAD, is the 14% corrected for inflammation? If so which methods of correction was used. 14% prevalence is considered ‘moderate VAD’

L74: What is the definition of “emerging regions”. where are they on a map? Please provide a map.

L77: 'onset of' is superfluous

L81: “therefore” is unnecessary

L83: 'Moreover' is unnecessary

L84: I suggest “problems”

STUDY SETTINGS AND DATA SOURCES

L92: Why are they called 'emerging'. Please explain.

SAMPLING PROCEDURES

L108: I presume you mean 'one' segment rather than 'only a'.

L110: 'A fixed number of' is redundant. The sentence can start with 'twenty-eight'

L112: What do the enumerators do if no-one was at home to be interviewed? or no children 6-23 within the household.

L117: Why female survey, were females the only targeted audience?

L118: Please be consistent with the use of mothers/caregivers

L119: The meaning of this sentence is unclear.

L121: How was that one child of twins selected? By ballot?

L122: ‘extracted’ is an odd way of expressing this

MEASUREMENTS OF VARIABLES

L130: Again, the reader can’t follow the augment clearly as you haven't defined what MNs and which preparations and recommended timeframes at the beginning

L131: add “/caregivers”

L137: Please clearly explain in background why deworming medication was included in this analysis MNs

L138: is this referring to any one of the recommended micronutrients, two or three?

L159: Do the national programs start providing deworming medication at 6 months of age rather than more normally from 12 months of age. What was the deworming medication?

L160: How reliable is the child health card as a source of information rather than the verbal history form the caregiver.

L164: Do you mean obstetric or history and/or parity and/or ante-natal care?

L168: These indicators measured to reach your definition of wealth need to be described.

L169: Are these ranks comparable to international definitions or just in relation to the cohort under investigation. Are you really comparing very poor, poor, less poor?

L170-1: This sentence is unclear.

DATA PROCESSING AND STATISTICAL ANALYSIS

L185: Again, do you mean parity/access to ANC?

RESULTS

L223: Again, were ALL the participants mothers of these children or were some caregivers?

OBSTETRIC HISTORY OF PARTICIPANT

L236: please define the term “multipara”

L237: acronym ANC not yet introduced and must be spelt in fill the first time it is used

L239: acronym PNC not yet introduced and must be spelt in fill the first time it is used

L241: Grand multipara must first be defined as mentioned above

CHILD CHARACTERISTICS AND COMMON CHILDHOOD ILLNESSES

L245: No definitions of average birth weight provided

L247: “were weighted” should be changed to 'were weighed'

L247: please insert % after these figures

L248: I presume you mean had 'either' rather than 'and'

L251: I suggest adding “within last 2 weeks” for diarrhoea, cough and fever which is I believe the timeframe under consideration

COMMUNITY LEVEL POVERTY, MEDIA EXPOSURE AND ACCESS TO HEALTH FACILITY

L254: ‘had had’ not ‘has’

L256: do you mean ‘was’ or ‘was not’

MICRONUTRIENT INTAKE AMONG CHILDREN 6-59 MONTH

L258: I don't think you adequately described your definition of minimum recommended in background or methods.

DISCUSSION

I think this section need to be carefully rewritten to ensure it relates to the findings in a systematic manner.

L361: discuss the association to “care” in these regions. like active response feeding for this age group since they are young compare to the older age group, who might be considered “can eat for themselves” in a family meal!!

CONCLUSIONS

L418-onwards are recommendations rather than conclusions based upon findings

REFERENCES

Placed ensure you comply with instructions to authors.

7. PLOS authors have the option to publish the peer review history of their article (what does this mean?). If published, this will include your full peer review and any attached files.

Reviewer #1: **Yes: **Anni-Maria Pulkki-Brännström

Reviewer #2: **Yes: **Habib Issa Kamara

Reviewer #3: No

---

## [Author Response · Author response to Decision Letter 2]

22 Dec 2020

Dear Sir/ Madam,

We appreciate and thank to the academic editor and reviewers for reviewing our manuscript for improvements. It is with great pleasure to receive the invaluable and constructive comments. 

We have incorporated all the comments and made necessary edition and corrections. All the comments with their point-by-point responses are included in the response to reviewers and we hope now we have addressed all the concerns. Therefore, we are kindly requesting you to review our revised manuscript.

---

## [Decision Letter · Decision Letter 3]

12 Feb 2021

PONE-D-20-13718R3

Micronutrient intake status and associated factors among children aged 6-23 months in the emerging regions of Ethiopia: a multilevel analysis of the 2016 Ethiopian demographic and health survey

PLOS ONE

Dear Dr. %Tsegaye Gebremedhin%,

Thank you for submitting your manuscript to PLOS ONE. After careful consideration, we feel that it has merit but does not fully meet PLOS ONE’s publication criteria as it currently stands. Therefore, we invite you to submit a revised version of the manuscript that addresses the points raised during the review process.

ACADEMIC EDITOR: Please insert comments here and delete this placeholder text when finished. Be sure to:

This manuscript is improving but would be benefit from another thorough review with a fluent English speaker with a background in public health. I have uploaded many commentsBe concise, to not repeat all finding in the text where they are clearly visible in the tables and are of no further importance to the discussion.Define minimum MN recommendations in the Abstract and again in MethodsProvide the multivariant findings in Abstract-ResultsIn discussion focus on the multivarient findings and their interpretationSelect the best references

We look forward to receiving your revised manuscript.

Kind regards,

Mary Hamer Hodges, MBBS MRCP DSc

Academic Editor

PLOS ONE

Additional Editor Comments (if provided):

This manuscript is improving with each revision. However it still needs substantial modifications before it is ready for publication in PLOSONE. I have made many detailed observations but the general messages are:

You have not clearly defined what the minimum recommendation of MNs intake is. Is it all 6 options, one of the six options or 3 of the six or something else?

In results be far more concise and after describing the cohort provide the coverage of all 6 of these MN options and refer the reader to the Tables for the other details unless there is something you especially want to draw their attention to and will be discussing later.

In discussion summarize the findings concisely before comparing with 1) other studies in Ethiopia and then 2) other studies in SSA.

You have far too many references for a study of this nature. I recommend to reduce to a maximum of 40 by picking only the best of the rest (most recent, most robust)

Focus on the mutlivariate analysis one at a time with your interpreation of their meaning. For example the most interesting observation was ANC (prenatal care) or distance to health centers.

Reviewers' comments:

Reviewer's Responses to Questions

**Comments to the Author**

1. If the authors have adequately addressed your comments raised in a previous round of review and you feel that this manuscript is now acceptable for publication, you may indicate that here to bypass the “Comments to the Author” section, enter your conflict of interest statement in the “Confidential to Editor” section, and submit your "Accept" recommendation.

Reviewer #2: All comments have been addressed

Reviewer #3: All comments have been addressed

2. Is the manuscript technically sound, and do the data support the conclusions?

Reviewer #2: Yes

Reviewer #3: Yes

3. Has the statistical analysis been performed appropriately and rigorously? 

Reviewer #2: Yes

Reviewer #3: Yes

4. Have the authors made all data underlying the findings in their manuscript fully available?

Reviewer #2: Yes

Reviewer #3: Yes

5. Is the manuscript presented in an intelligible fashion and written in standard English?

Reviewer #2: No

Reviewer #3: Yes

6. Review Comments to the Author

Reviewer #2: Abstract- The author has not been too clear with his conclusion. He says ‘The micronutrient intake status in the study population in this area was low compared to the national recommendation. Promoting vitamin, A and iron-rich foods and micronutrient powders are better for micronutrient enrichments. Strengthen supplementations 35 and deworming alongside the community-based maternal and child health services would 36 improve micronutrient intake among children. There are a couple of typos here and grammatical errors that need to be addressed to make the conclusion concise. The author could try and rephrase the conclusion to read: The status of micronutrient intake among the study population was low when compared to the national recommended threshold. Strengthening supplementation and deworming in addition to maternal and child health services would improve micronutrient intake among children

Line 141-142: In the sampling procedure, the author has not explained why mothers with twins were only interviewed for one child. The author could consider explaining this sampling method for easy comprehension of sampling at the household for readers.

Line 285-286: This sentence does not read well. The author could rephrase the sentence to read- Of those who took the recommended micronutrient, only 12.9% had received three or more types of micronutrients

Line 293-294: There are a few typos in this sentence: “Around 47% (95% CI: 44.1-50.3) of the children 293 received vitamin A supplements in the six months preceding the interview. Besides, 8.4% (95% 294 CI: 5.5-8.7) of the children aged 12 to 23 months of were received deworming medication in the six months preceding the interview (Table 4).” The author could rewrite to make it easier for readers to understand what he is trying to say.

Line 361-362: Vitamin A supplement in Nigeria was 45% which is comparable with the current finding [53]. However, our study differs from the finding in India (30.4%) [54]. I suggest the author sticks with studies whose results are comparable to this study. Only compare finding in Nigeria to this and take out the study in India.

Line 363: This study's result is much lower than that of the national target of over 90% [55]. The way this sentence is written is not grammatically sound. The author could consider re-writing to read: “Results from this study shows much lower micro nutrients uptake compared to the national target of over 90%

Line 372, 388-391, 398-399 either are not writing in proper English or are not clear. The author need to rewrite again to make them readable and clear to the reader. I have made some suggestions here. Instead of saying micronutrient intake among children whose mothers worked in agriculture was higher than children whose mothers did not have work; Rephrase sentence to read – Mothers who were housewives or who had no formal/paid jobs

Replace Also, even if vitamin A supplement is effective for 6-11 months, especially when used with the vaccine against measles…. With when administered with measles vaccines

Instead of saying Furthermore, they might learn about the value of iron intake that supplements, the author could re-write sentence to read: caregivers may have learned or acquired knowledge of iron supplements during their ANC follow-up.

Line 412-413: Again, instead of saying” Besides, since the latter two regions have dense forests and water reservoirs, they could get wild fruit and fish, which are good micronutrients” the author should rephrase as “Besides, since the latter two regions have dense forests and water reservoirs, caregivers could get wild fruit and fish, which are good sources of micronutrients”

I recommend the author to read the whole manuscript again and ensure consistency in presenting results. Adjust all percentages to 1 decimal point.

Once more I suggest the author gets a native English speaker, to proof read the manuscript and ensure it is written in clear, simple English language.

Reviewer #3: Abstract

L14:I suggest you add "include" before the word food

Conclusion

L 33: What is the national recommended threshold?

L 34: Supplementation of what?

Introduction

L 75: This target should be sighted in the abstract above, L33, of my previous comment

Measurement of Variables

L 174 : Were the mothers shown a sample of the sachet during the interview?

L 182: I suggest including the word "health" before the word "card"

Micronutrient intake status among children aged 6-23 months

L 294: I suggest deleting "of were"

7. PLOS authors have the option to publish the peer review history of their article (what does this mean?). If published, this will include your full peer review and any attached files.

Reviewer #2: **Yes: **Habib Issa Kamara

Reviewer #3: No

---

## [Author Response · Author response to Decision Letter 3]

23 Mar 2021

Dear Editor and Reviewers,

We are very much thankful for your constructive comments and suggestions, which are really useful for the improvement of our manuscript. We have addressed all your concerns and the responses are included in the 'Response to Reviewers point-by-point' and clean version of the revised manuscript and submitted for your revision. 

Thank you in advance!

---

## [Decision Letter · Decision Letter 4]

26 Apr 2021

PONE-D-20-13718R4

Micronutrient intake status and associated factors among children aged 6-23 months in the emerging regions of Ethiopia: a multilevel analysis of the 2016 Ethiopian demographic and health survey

PLOS ONE

Dear Dr. %Tsegaye Gebremedhin%,

Thank you for submitting your manuscript to PLOS ONE. After careful consideration, we feel that it has merit but does not fully meet PLOS ONE’s publication criteria as it currently stands. Therefore, we invite you to submit a revised version of the manuscript that addresses the points raised during the review process.

ACADEMIC EDITOR:

Please pay careful attention to the reviewers comments on language and use to abreviations and references

We look forward to receiving your revised manuscript.

Kind regards,

Mary Hamer Hodges, MBBS MRCP DSc

Academic Editor

PLOS ONE

Journal Requirements:

Reviewers' comments:

Reviewer's Responses to Questions

**Comments to the Author**

1. If the authors have adequately addressed your comments raised in a previous round of review and you feel that this manuscript is now acceptable for publication, you may indicate that here to bypass the “Comments to the Author” section, enter your conflict of interest statement in the “Confidential to Editor” section, and submit your "Accept" recommendation.

Reviewer #2: (No Response)

2. Is the manuscript technically sound, and do the data support the conclusions?

Reviewer #2: Partly

3. Has the statistical analysis been performed appropriately and rigorously? 

Reviewer #2: Yes

4. Have the authors made all data underlying the findings in their manuscript fully available?

Reviewer #2: Yes

5. Is the manuscript presented in an intelligible fashion and written in standard English?

Reviewer #2: No

6. Review Comments to the Author

Reviewer #2: Abtract, conclusion

L35: Although you have now clearly defined the sources of MN you have not defined the national recommendations. is it that all individuals should have all of these interventions?

Introduction

L42:Include (there are others)

L48: The last part of this sentence needs amending

L50: Instead of saying vitamins and minerals why not stay with MN?

L51: At the beginning of this sentence it would be good to state which year(s) you are referring to.

L72: Please state whether this was corrected for inflammation and if so by which method. L75: Rather than 'on top of' I suggest 'in addition to'

L89: Problem (no s)

L131: 'Those second' do you mean 'those years old and older'?

L133: For mother /caregivers with twins only one was selected by convenience

L183: You mean 9 instead of an television?

L240: Having defined grand multi para please use it here, unless there was a discrepancy between giving birth and living children.

L247: I recommend reducing this text further by referring the reader to the table. Only points of interest in the final analysis need a mention which does not include average birth weight and diarrhoea. However, the final analysis does include mothers occupation etc.

L251: In the text you are using the word weight and here you are referring to size. Please clarify again as earlier you actually specified weight in kg whereas previously, I thought this was a subjective assessment by the mother/caregiver.

L259: Rather than 'Moreover' use 'Only'

L261: Better to continue the sentence rather than break it with 'additionally'

L286: Rather than antenatal visit stay with ANC

L300: I believe the manuscript would make a bigger impact if you limit this table to only those findings of statistical significance. The text could then simply stat that the other characteristics were not significant.

Discussion

L312: I see you have dropped the child’s weight now but there is still confusion over whether you were recording weight or size. Add had consumed

L313: Do you mean or iron? No need for the word 'reports'

L315: No need to start this sentence with 'besides'

L316: Got, not get.

L318: Current result? are you comparing EDHS 2011 with 2016 or with this analysis?

L322: Use MN

L326: Rather than 'the current finding' I suggest 'this study'

L329: No need for 'in Ethiopia'

L331: The sentence starting on L331-332 seems redundant although the citations might be useful

L332: Again, starting the sentence with 'on the other way' is redundant.

L334: What do you mean by low ability?

L338: The sentence starting on L336-339 does not make sense. Why would children not be eligible for VAS

L344: Another explanation might be that mothers with follow-up live nearer to health facilities of have more time/money available to attend ANC.

L347: Same comment as above

L351: A few

Strength and limitations of the study

L368: No need to expand this sentence. You can stop after factors.

L369: No need to start the sentence with 'besides'

L372: The last sentence in this para can be simplified to 'the DHS methodology allows for comparison with other settings'.

L373: This sentence is redundant as it is neither a strength not a weakness.

L375: This second sentence can be removed as you are not assessing adequacy only the MN intake of MN recommendations.

L376: No need to start with 'furthermore'

L377: Here you are using VAS but not in previous sections. Once an abbreviation has been introduced please ALWAYS use it.

L378: I don't feel the last sentence is required.

Conclusion

L381: Conclusion should be only 2-3 sentences just starting the essential facts not going over the details again. Some of what you say here could be in the discussion (but not all)

References

L4023: Please review these citations and ensure only those of importance are used and that none are used twice.

7. PLOS authors have the option to publish the peer review history of their article (what does this mean?). If published, this will include your full peer review and any attached files.

Reviewer #2: **Yes: **Habib Issa Kamara

---

## [Author Response · Author response to Decision Letter 4]

11 Jul 2021

Dear editor and reviewer,

Thank you for your valuable comments, suggestions and insights which really improved our manuscript. All the responses are included in the "Response to reviewers' file.

---

## [Editor Report · Decision Letter 5]

18 Aug 2021

PONE-D-20-13718R5

Micronutrient intake status and associated factors among children aged 6-23 months in the emerging regions of Ethiopia: a multilevel analysis of the 2016 Ethiopia demographic and health survey

PLOS ONE

Dear Dr. %Tsegaye Gebremedhin%,

Thank you for submitting your manuscript to PLOS ONE. After careful consideration, we feel that it has merit but does not fully meet PLOS ONE’s publication criteria as it currently stands. Therefore, we invite you to submit a revised version of the manuscript that addresses the points raised during the review process.

Please address the comments on wealth and birth weight/size and consider deleting Table 3 and just making a statement that none of those factors were found to be of significance with regard to MN intake.

We look forward to receiving your revised manuscript.

Kind regards,

Mary Hamer Hodges, MBBS MRCP DSc

Academic Editor

PLOS ONE

Journal Requirements:

Additional Editor Comments (if provided):

You have made some improvements but there is still too much information being presented in Tables 2 and 3 that have no relevance to your final analysis. Confusion over how you have defined wealth status/index. quintile/quantile and birth weight/size remain. If data in tables 2 and 3 have been analyzed and found 'Not Significant' I suggest to reduce the amount of data being presented to the reader so that attention can be drawn to important findings: mothers occupation, ANC visits, child's age, residence and region.

Please define what the governments recommendations for minimum dietary diversity are in terms of food groups consumed within last 24 hours. Below I have drawn attention to some minor typos and presentation issues

L 14: rather than ‘prevent’ I believe these recommendations/interventions help to ‘reduce’.

L 14: I think this sentence need to be modified as diets have been ‘promoted’ and programs for the distribution MNPs or others supplements for iron and VA and deworming ‘have been implemented’.

Methods

L 21: ‘consumed within the previous’ 24 hours

L 22: ‘supplementation with the previous’ seven days

Result

L 31: This sentence seems unfinished. Are these factors that increase MN uptake or decrease MN intake, or mixed some increased and some decreased? Same comment for last sentence.

Conclusions L 35: recommendation’s’. it is still not clear what the Government recommendations are whether it is a diet with VA or iron rich foods plus VAS or iron supplementation or MNPs. I suspect you are trying to say diet rich (in VAS or iron) or VA or Iron supplementation or MNPs but it is still not clarified in Background.

Introduction

L 45: May lead would be better replaced by 'contribute to'

L 64: 6-59 months semi-annually in a (no need for ‘by’)

L 65: Which ones? Iron folic acid (IFA) or multiple micronutrient supplements (MMS)?

L 69: Were receiving ‘a diet of minimal dietary diversity’ Please don’t use an abbreviation such as MDD without first introducing it. What does the government recommend for MDD? Three or more of the food groups as elaborated further in Results? Please specify.

L 82: Hotspot’s’ (pleural)

L 85: I think the last section of this sentence ‘that exacerbate MN deficiency’ could be deleted. These factors to not exacerbate MN deficiency they limit the ability to address the issue through interventions.

L 88: Finding’s’

L 89: Problem’s’

Study setting and data source

L 104: ‘and access to health and education services’

Measurements of variables

L 143: ‘consumed within the previous seven days’

L 144: ‘the previous’ six months

L 170: reviewers have asked previously how birth weight/size was assessed. Was it mothers recall of size or an actual weight taken at birth and recorded in a child health card? If it was mothers recall them please use large, average or small as reported in table 3.

L 171: Where did you get this classification? 4kgs seem excessive. please double check and provide a reference. As the results of this weight/size was not fund to be of any significance it might be best to omit it altogether to avoid these critisims.

L 172: This section lines 172-187 is unclear. Sometime you use quintile, sometimes quantile and sometimes index. On line 175 you have 3 subdivisions and on line 181 you have 5 (usual for quintiles). please clarify.

L 174: Quintile or index?

Result

L 231: What do to mean here wealth quintiles?

L 241: Why have you written this in words rather than given the actual figure 56.4%?

L 245 Table 2: What is the merit or presenting this data as it is not used in the further analysis or found significant. What is the merit of presenting all this date? It is not used or found significant in the analysis

What is the merit of presenting this data if it was not further analysis or found significant? A reader interested in these figures can find them in the EDHS. Only data that directed the reader to your important findings really deserves attention.

L 252: The merit of Table 3 is questionable. None of these factors were associated with MN intake.

L 253: Date that was not found to be of significance in the analysis could be removed from these tables to draw attention to elements there are of significance.

If the birth interval or birth order had no significant association with MN intake what is the merit of presenting the date?

what is the merit of the current child weight being presented if it is not related to the child age (W/A) or height (WH)? If you are not using this data in the analysis, please delete both these rows.

These last 3 section on diarhoea, cough and fever get not further mention in the analysis. If they were not significantly assassinated with MN intake, please delete these 6 rows and jut make a statement that there was not associated with MN intake.

Community level variables

L 256: Past tense 'were'

L 257: Now you are using status as opposed to index, quintile/quantile.

L 259: Quantile needs a better explanation in methods

Individual and community level factor of micronutrient intake status (fixed effects)

L 293: Here you could list all the factors that were not significant.

L 301: Versus urban residents

Discussion

L 316-322 you are inconsistent with the use of decimal places sometime non and sometimes 2. For example: 38% and 43.17%. This is distracting for the reader.

L 323: The term 'housewives' has not previously been used.

L 347: Citing the author is not the PLOS ONE format for referencing

Strength and limitation of the study

L 367: It is normal practice to just draw attention to the study limitations as the last paragraph in discussion (without a sub-heading)

Conclusion

L 375: You have not clearly defined the national dietary recommendations (number of food groups to be consumed in previous 24 hours). For deworming it might be >75% in the last 6 months. But what about the other options?

L 377: This second sentence is not based upon findings.

L 379-381: We have only cited deworming and MNPs. What about iron and VAS?
---

## [Author Response · Author response to Decision Letter 5]

28 Sep 2021

Dear editor, 

Thank you for your comments and suggestions that improved our manuscript. All responses to the editor's comments and suggestions are included in the point by point response letter and finally we have submitted the Responses to Reviewers letter, clean version of the revised manuscript and the marked copy documents. We are kindly requesting to review our revision.

Thank you!

---

## [Editor Report · Decision Letter 6]

11 Oct 2021

Micronutrient intake status and associated factors among children aged 6-23 months in the emerging regions of Ethiopia: a multilevel analysis of the 2016 Ethiopia demographic and health survey

PONE-D-20-13718R6

Dear Dr. %Tsegaye Gebremedhin%,

We’re pleased to inform you that your manuscript has been judged scientifically suitable for publication and will be formally accepted for publication once it meets all outstanding technical requirements.

Kind regards,

Mary Hamer Hodges, MBBS MRCP DSc

Academic Editor

PLOS ONE

Additional Editor Comments (optional):

This is much improved and not worthy of publication. The abbreviation (MN) need NOT be introduced in the abstract since you do not use it again for the rest of the abstract.
---

## [Editor Report · Acceptance letter]

14 Oct 2021

PONE-D-20-13718R6 

Micronutrient intake status and associated factors among children aged 6-23 months in the emerging regions of Ethiopia: a multilevel analysis of the 2016 Ethiopia demographic and health survey 

Dear Dr. Gebremedhin:

I'm pleased to inform you that your manuscript has been deemed suitable for publication in PLOS ONE. Congratulations! Your manuscript is now with our production department. 

Kind regards, 

on behalf of

Dr. Mary Hamer Hodges 

Academic Editor

PLOS ONE